# Impacts of ambient air quality on acute asthma hospital admissions during the COVID-19 pandemic in Oxford City, UK: a time-series study

Ajit Singh [1,2] Gabriella L Morley,[3] Cécile Coignet,[4] Felix Leach [5] Francis D Pope,[1] Graham Neil Thomas,[2] Brian Stacey,[6] Tony Bush,[5,7] Stuart Cole,[8] George Economides,[8] Ruth Anderson,[8] Pedro Abreu,[9] Suzanne E Bartington [2]

AS and GLM are joint first authors.

For numbered affiliations see end of article.

**Correspondence to**
Dr Suzanne E Bartington;
s.bartington@bham.ac.uk

## ABSTRACT

**Objectives** The study aims to investigate the short-term associations between exposure to ambient air pollution (nitrogen dioxide ($NO_2$), particulate matter pollution—particles with diameter<2.5 µm ($PM_{2.5}$) and $PM_{10}$) and incidence of asthma hospital admissions among adults, in Oxford, UK.

**Design** Retrospective time-series study.

**Setting** Oxford City (postcode areas OX1–OX4), UK.

**Participants** Adult population living within the postcode areas OX1–OX4 in Oxford, UK from 1 January 2015 to 31 December 2021.

**Primary and secondary outcome measures** Hourly $NO_2$, $PM_{2.5}$ and $PM_{10}$ concentrations and meteorological data for the period 1 January 2015 to 31 December 2020 were analysed and used as exposures. We used Poisson linear regression analysis to identify independent associations between air pollutant concentrations and asthma admissions rate among the adult study population, using both single ($NO_2$, $PM_{2.5}$, $PM_{10}$) and multipollutant ($NO_2$ and $PM_{2.5}$, $NO_2$ and $PM_{10}$) models, where they adjustment for temperature and relative humidity.

**Results** The overall 5-year average asthma admissions rate was 78 per 100 000 population during the study period. The annual average rate decreased to 46 per 100 000 population during 2020 (incidence rate ratio 0.58, 95% CI 0.42 to 0.81, p<0.001) compared to the prepandemic years (2015–2019). In single-pollutant analysis, we observed a significantly increased risk of asthma admission associated with each 1 µg/m³ increase in monthly concentrations of $NO_2$ 4% (95% CI 1.009% to 1.072%), $PM_{2.5}$ 3% (95% CI 1.006% to 1.052%) and $PM_{10}$ 1.8% (95% CI 0.999% to 1.038%). However, in the multipollutant regression model, the effect of each individual pollutant was attenuated.

**Conclusions** Ambient $NO_2$ and $PM_{2.5}$ air pollution exposure increased the risk of asthma admissions in this urban setting. Improvements in air quality during COVID-19 lockdown periods may have contributed to a substantially reduced acute asthma disease burden. Large-scale measures to improve air quality have potential to protect vulnerable people living with chronic asthma in urban areas.

### STRENGTHS AND LIMITATIONS OF THIS STUDY

⇒ This study investigated the association between ambient air pollution exposure and risk of admission to hospital for asthma.

⇒ The study includes an at-risk population of >800 000 person-years.

⇒ Associations were evaluated using single and multipollutant models.

⇒ Causation cannot be inferred from this observational study.

⇒ A single city study may limit wider generalisation of findings.

## INTRODUCTION AND BACKGROUND

Asthma is a non-communicable chronic disease characterised by airway hyper-responsiveness, reversible airflow obstruction and inflammation.[1] As the most common respiratory condition worldwide, asthma affects more than 300 million people globally including 12% of the population in the UK[2] with 4.3 million UK adults requiring treatment for the condition.[3] In 2019, asthma contributed to 368 years lived in disability (95% CI 238 to 546) and 410 per 100 000 Disability Adjusted Life Years (DALYs) (95% CI 280 to 588)[4] among the UK population, with the greatest disease burden occurring in areas of high deprivation.[3] Asthma can also be fatal and approximately 1200 people a year die from asthma in the UK.[5] As there is no permanent cure, clinical management is directed towards preventing or reducing the severity of exacerbations and controlling symptoms including wheeze, cough and breathlessness.[6 7] Primary prevention of asthma morbidity and mortality is therefore of major public health significance.

Asthma has multifactorial aetiology and is likely the result of complex gene and

environmental interactions.[8] Environmental triggers may exacerbate symptoms so chronic asthma patients are advised to modify lifestyle and behaviours to reduce their exposure as a method for preventing exacerbations, alongside the use of inhaled or oral medication.[6] In recent years, there has been increasing concern regarding the contribution of poor air quality to disease morbidity and mortality. Almost all of the world's population live in areas which do not meet 2021 WHO Global Air Quality Guidelines.[9–11] In the UK, pollutants of major health concern include nitrogen dioxide ($NO_2$) which can cause airway inflammation, and fine particulate matter pollution—particles with diameter <2.5 µm ($PM_{2.5}$), which can enter the circulation, with long-term exposure to both pollutants linked to cardiorespiratory disease.[12–15] People with existing chronic respiratory diseases such as asthma or chronic obstructive pulmonary disease are more vulnerable to $PM_{2.5}$ particle and $NO_2$ exposure[16] thereby exacerbating existing health inequalities. Long-term exposure to fine $PM_{2.5}$ and $NO_2$ is recognised to be causally associated with increased risk of developing child and adult asthma.[17 18] Mechanisms linking air pollution exposure to asthma include modification of inflammatory pathways, leading to airway remodelling and enhancement of sensitivity to environmental allergens.[19] A recent meta-analysis of 22 studies undertaken in 12 countries showed a significant association between ambient air pollution exposure ($NO_2$, $PM_{2.5}$, CO and $O_3$) and risk of asthma exacerbations among adults and children. A 5-year time-series study (2009–2013) across primary care settings in London identified that an increase in short-term exposure to $NO_2$ and $PM_{10}$ was associated with a significant increase in respiratory consultations, inhaler prescriptions or both.[20]

Improvements in ambient air quality have also been linked to reduced asthma morbidity. Implementation of emergency public health measures in response to the COVID-19 pandemic led to substantive reductions in industrial and transportation activities from spring 2020 and related changes in air pollution levels.[21–23] Studies in major cities worldwide indicated lockdown periods were associated with a reduction in major air pollutant ($NO_2$, $PM_{10}$ and $PM_{2.5}$) concentrations.[24–29] Studies in the UK revealed a similar pattern to global trends in terms of air quality changes,[22 24 30] most notably $NO_2$ reductions in urban areas.[21]

The outbreak of COVID-19 led to concerns that this viral pandemic would increase the number of asthma exacerbations, resulting in an influx of emergency department visits and hospitalisations among asthma patients. However, a national analysis of anonymised patient records in Scotland and Wales identified substantial reductions in severe asthma exacerbations leading to hospital admission during COVID-19 lockdown periods, although mortality rates remained unchanged.[31] A reduction in attendance to primary care for asthma exacerbations was also identified in a large-scale analysis of routine data in England.[32] Similar reductions in asthma emergency attendances and admissions among children were

also identified across 15 countries worldwide,[33] suggested to be due to reduced exposure triggers and increased treatment adherence. While the trends described in air pollution are replicated in smaller cities and towns, the majority of existing research has been conducted in major cities[24 30] with little opportunity to understand the relationship between changes in ambient air quality and asthma admissions among adults in smaller towns and cities in which 45% of the UK population live.[34]

In this study, a time-series analysis was conducted to investigate the association between ambient air pollution exposure and risk of unplanned (emergency) admissions among adults with asthma living in Oxford, UK.

## MATERIALS AND METHODS

This study is part of the Natural Environment Research Council (NERC) funded OxAria Study that aims to understand the impacts of COVID-19 on air and noise quality and associated health impacts in Oxford City (https://oxaria.org.uk/).

### Study design

This study uses a retrospective time-series ecological design at the city level. The unit of analysis was a monthly count of the hospital admissions for adults, admitted with a primary diagnosis of acute asthma, during the study period of 1 January 2015–31 December 2020. Acute asthma is a serious condition involving progressive worsening of asthma symptoms, including breathlessness, wheeze, cough and chest tightness. Patients with severe of life-threatening acute asthma require immediate hospital treatment.

### Setting

Oxford is a small historical city with an area of 46 km$^2$ and 68 m above sea level.[35] Oxford has a warm and temperate climate with mean annual temperature 10.3°C, mean annual cumulative rainfall 708 mm. The diverse population of this international city is approximately 152 000, of which 79.6% are adults.[34] The city has two universities with approximately 34 000 students enrolled. The city is a major centre for employment with a thriving economy, with an estimated 46 000 people usually coming to the city for work, on a daily basis, before the COVID-19 pandemic.[36] The population is served by Oxford University Hospitals National Health Service (NHS) Foundation Trust and comes under the jurisdiction of the Oxfordshire Clinical Commissioning Group (OCCG). A recent study by the OxAria team provides a detailed geographical, meteorological and environmental description of the study setting.[22]

### Study population

The study population is adults aged >18 years living within Oxford City postcode areas OX1–OX4 (see online supplemental figure S1).

## Data collection

### Health outcome data

Hospital Episode Statistics (HES) data were accessed through OCCG. The study sample was obtained from weekly aggregated HES admissions data to a single tertiary hospital (John Radcliffe Hospital, Oxford) where all patients admitted with a primary diagnosis of asthma, as the outcome of interest, were included. Cases were extracted based on the inclusion criteria of age (>18 years), residential postcode district (OX1–OX4) and Clinical Classifications identifier: ICD-10 (International Classification of Diseases 10th Revision code) coding for acute asthma (ICD-10, J45–J46).[37] Exclusion criteria were all cases of wheeze (ICD-10, R06.2), cough (ICD-10, R05) and influenza (ICD-10, J11). Missing values within the HES outcome dataset were minimal (<4%) over the total study duration. Here, missing values where present were replaced by interpolation based on the adjacent values. The mean monthly asthma hospital admissions rate was calculated and used as the unit of analysis.

### Air pollution exposure and weather data

Hourly air pollutant concentrations ($NO_2$, $PM_{2.5}$ and $PM_{10}$) for the period 1 January 2015 to 31 December 2020 were obtained at the Automatic Urban and Rural Network St Ebbe's urban background site (UKA00518),[38] located centrally in the OX1–OX4 postcode area (covers 3–5 miles radius). Monthly mean air pollution concentrations were calculated from hourly datasets and analysed for the study period.

Hourly meteorological data including relative humidity (RH) and air temperature (T) from 1 January 2015 to 31 December 2020 were obtained from Oxford Radcliffe Observatory[39]—an urban background site, which was located in the OX2 postcode area. The proportion of missing hourly data was 16% ($NO_2$), 20% ($PM_{10}$) and 24% ($PM_{2.5}$) across the 6-year study duration. This was not replaced with imputed data given the potential for fluctuations over this time period which may result in an inaccuracy or bias.

### COVID-19 lockdown classification

Data were obtained from 1 January 2015 to 31 December 2020. We consider two national lockdown periods in England in 2020: (1) lockdown 1 (23 March 2020–15 June 2020) and (2) lockdown 2 (5 November 2020–2 December 2020).[22 23]

### Statistical analysis

Mean annual air pollutant concentrations ($NO_2$, $PM_{2.5}$, $PM_{10}$) for the study period (2015–2020) were calculated and compared with WHO guideline levels.[11] Correlation between air pollutant concentrations and meteorological variables (air temperature and RH) was explored using Pearson correlation matrices and coefficient.

The incidence rate of asthma admissions was determined for each year and the incidence rate ratio (IRR) compared with 2019 (baseline year), with a comparison of 2020 admissions rate to a 5-year average by Poisson regression analysis. A Pearson correlation matrix was used to compare asthma admissions and air pollutant concentrations for 2020 and prepandemic years (2015–2019). To compare the trends between asthma admissions and air pollutant concentrations, values normalised by the mean were calculated and a time series was generated.

Poisson linear regression analysis was used to quantify independent associations between monthly air pollutant concentrations and incidence of asthma admissions among the adult study population, using single ($NO_2$, $PM_{2.5}$, $PM_{10}$) and multipollutant ($NO_2$ and $PM_{2.5}$, $NO_2$ and $PM_{10}$) models, with adjustment for temperature and RH. All statistical analyses were conducted in STATA V.15 applying 95% CIs and two-tailed p values at <0.05 considered statistically significant.[40]

### Patient and public involvement

Public members were involved in the design and conduct of this study including public representation on the study steering committee.

## RESULTS

### Air quality

Overall mean annual average $NO_2$ and $PM_{2.5}$ concentrations for the total study duration exceeded the WHO Global Air Quality guideline average levels of 10 µg/m³ and 5 µg/m³, for all study years, respectively. A reduction in annual average $NO_2$, $PM_{10}$ and $PM_{2.5}$ concentrations (0.74, 0.42 and 0.97 µg/m³ per year, respectively) was observed from 2015 to 2019 which further reduced in 2020 (figure 1A). There was also evidence of seasonal peaks with higher pollutant levels in the winter months

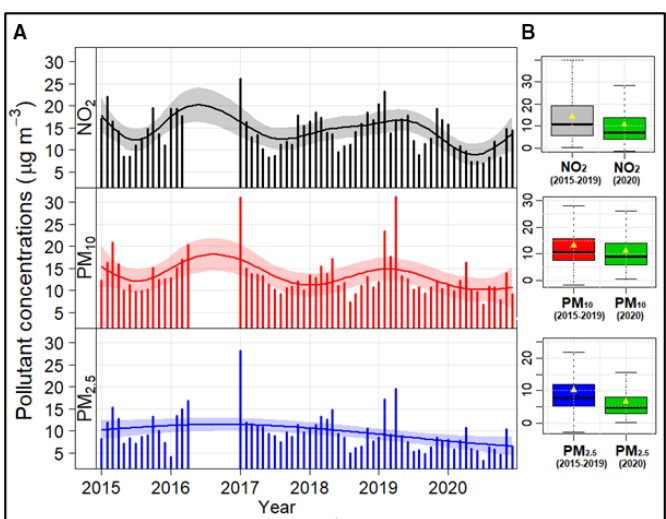

**Figure 1** (A) Time series (2015–2020) of monthly mean observed pollutant ($NO_2$, $PM_{10}$ and $PM_{2.5}$) concentrations at urban background site (St Ebbe's) in Oxford, where shaded lines represent the smooth fit at the 95% CI. (B) Box plots of the daily averaged air pollutant concentrations in 2020 vs 2015–2019. $NO_2$, nitrogen dioxide; $PM_{2.5}$, particulate matter pollution—particles with diameter <2.5 µm.

for all three pollutants. The significant reductions in air pollutant concentrations in 2020 compared with 2015–2019 were attributed to COVID-19 national lockdown measures, primarily due to lower traffic emissions (figure 1B).[22]

It is observed that the mean $NO_2$ concentration reduced by 26.7% to $10.7\,\mu g/m^3$ in 2020 (95% CI 8.7 to $12.7\,\mu g/m^3$), which was significantly (p<0.005) lower than the 5-year average (2015–2019) concentration of $14.6\,\mu g/m^3$ (95% CI 12.6 to $16.7\,\mu g/m^3$). Similarly, the mean $PM_{10}$ concentrations significantly (p<0.05) reduced by 18.6% in 2020 to $10.8\,\mu g/m^3$ (95% CI 9.1 to $12.4\,\mu g/m^3$) compared with $13.2\,\mu g/m^3$ (95% CI: 11.2 to $15.2\,\mu g/m^3$) from 2015 to 2019. Relatively, $PM_{2.5}$ levels showed a greater reduction than the other two pollutants, with mean $PM_{2.5}$ concentrations significantly (p<0.005) falling by 33.5% to $6.7\,\mu g/m^3$ (95% CI 5.3 to $8.1\,\mu g/m^3$) in 2020 from $10.1\,\mu g/m^3$ in 2015–2019 (95% CI 8.4 to $11.8\,\mu g/m^3$). However, a recent study using a machine learning for deweathering and detrending analysis has shown that these reductions were not only due to a reduction in emissions but also influenced by the weather.[22]

$PM_{2.5}$ and $PM_{10}$ levels were strongly positively correlated in both 2015–2019 (0.96) and 2020 (0.94) (online supplemental figure S2). A weakly positive correlation was also observed for $NO_2$ and $PM_{2.5}$, higher temperatures were associated with lower levels of air pollution, with a negative correlation between all temperature and all three pollutants although this was weaker in 2020. RH had a weakly positive association throughout the study period.

## Asthma admissions

The overall incidence of asthma admissions for the total study period was 7.8 per month per 100 000 population. This varied by season, with the highest admissions rate during winter and spring, and the lowest in the summer months (online supplemental figure S3). We found that there were two notable peaks in admissions of adults with asthma in late 2016 and 2019 that coincided with peaks in PM concentrations (see figure 2). Overall, the asthma

**Table 1** Annual incidence of adult hospital admissions for acute asthma by year (2015–2020), compared with 2019 as baseline

| Year | Annual incidence rate* | IRR | 95% CI | P value |
| --- | --- | --- | --- | --- |
| 2015 | 62.78 | 0.83 | 0.61 to 1.11 | >0.05 |
| 2016 | 79.00 | 1.04 | 0.79 to 1.38 | >0.05 |
| 2017 | 91.59 | 1.20 | 0.92 to 1.58 | >0.05 |
| 2018 | 84.20 | 1.12 | 0.84 to 1.46 | >0.05 |
| 2019 | 76.08 | Baseline | | |
| 2020 | 45.70 | 0.60 | 0.43 to 0.82 | <0.01 |

*Per 100 000 population.
IRR, incidence rate ratio.

admissions rate in 2020 was 42% lower than the 5-year average (IRR 0.58, 95% CI 0.42 to 0.81, p<0.001) (online supplemental table S1). There was no difference between the incidence of admissions in 2015–2018 compared with 2019, while there was a significantly reduced risk of admission in 2020 (IRR 0.6, p<0.01) compared with 2019 (table 1).

Further, figure 2 shows the monthly count of hospital admissions of adults with asthma in 2020 and compares with the monthly average pre-epidemic years (2015–2019). We found that the number of admissions for the whole first pandemic year after March 2020 was lower than the 5-year average. Although, there was a similar pattern in summer 2020 admissions compared with the 5-year average summer, where the admission rate remained lower. In contrast, a higher peak was noted in March 2020 compared with the previous 5 years. We also observe a peak in admissions during September, with admission rates nearly matching the 5-year average. There was a marked decline in admissions during the preceding months in 2020 which was not seen during the prepandemic years.

### Understating the effect of air quality on asthma admissions in adults in response to COVID-19 measures in Oxford

Monthly mean normalised time series of air pollutant ($NO_2$, $PM_{10}$ and $PM_{2.5}$) concentrations and asthma admissions are presented in figure 3. Overall, variation in asthma admissions among adults mostly followed trends in air pollution over the 6-year study period, where a positive association between air pollutants and asthma admissions was observed in 2015–2019 with a relatively stronger correlation than in 2020 (figure 3 and online supplemental file). However, there was a negative correlation between air temperature and asthma admissions in 2020 and whereas a positive correlation was observed in prepandemic years (online supplemental table S2).

Further, to understand the association between air pollutants and asthma admissions among adults,

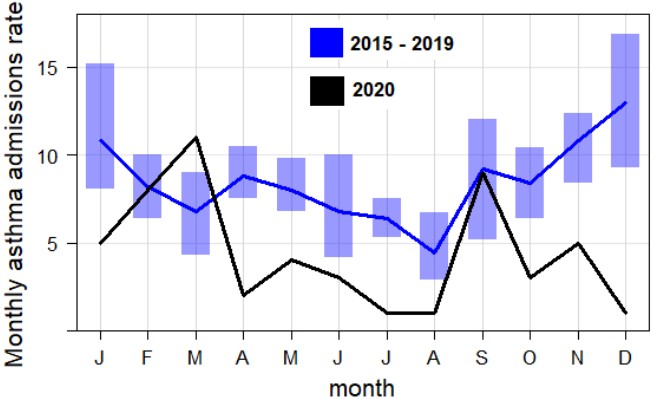

**Figure 2** Mean monthly rate of adult asthma admissions in Oxford asthma in 2020 vs a 5-year average. The shaded boxes represent the 95% CI.

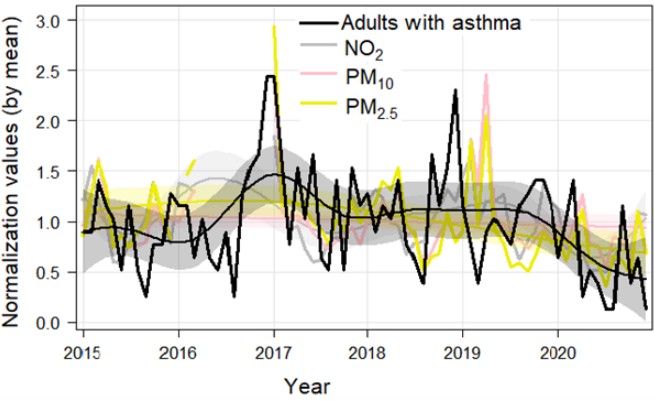

**Figure 3** Normalised time series of monthly asthma admissions with air pollutants during 2015–2020. The shaded lines represent the smooth fit at the 95% CI. $NO_2$, nitrogen dioxide; $PM_{2.5}$, particulate matter pollution—particles with diameter <2.5 µm.

single-pollutant and multipollutant models were generated applying Poisson regression analysis (table 2). In the single-pollutant model, which was adjusted for temperature and RH during the study period, showed that the risk of asthma admission increased significantly with increasing concentrations of $NO_2$ and $PM_{2.5}$. This was a 4% increase in risk for every 1 µg/m³ increase in mean monthly $NO_2$ and an approximately 3% increase in risk for every 1 µg/m³ increase in mean monthly $PM_{2.5}$. For every 1 µg/m³ increase in mean monthly $PM_{10}$, there was also a 1.8% increase in the risk of asthma admission, however, this failed to reach significance despite narrow CIs.

Within the multipollutant model, the effect of each individual pollutant was non-significant. A weak correlation between pollutants makes it difficult for the model to separate the effects of correlated pollutants. Overall, in the multipollutants model, we identified a non-significant 2.6% increase in risk of asthma admission per month predicted for every 1 µg/m³ increase in $NO_2$ and a non-significant 1.7% increase in risk of admission for every 1 µg/m³ increase $PM_{2.5}$, which approximates the risk for $PM_{10}$.

## DISCUSSION

This study examines for the first time the effects of ambient air quality on asthma-related hospital admissions in adults in Oxford city (UK), including impacts of the COVID-19 pandemic. A positive association was observed between risk of emergency asthma admission and air pollutant levels in Oxford, with hospital admission risk significantly increased even with modest increases in mean monthly $NO_2$ and $PM_{2.5}$ concentrations in single-pollutant models. We also found a 42% reduction in the risk of acute asthma admission among adults living in Oxford in 2020 compared with the 5-year average, suggesting air quality improvements during COVID-19 lockdown periods may have contributed to a reduction in severe asthma exacerbations. However, this observed association is in the context of a general reduction in emergency admissions for all causes during the COVID-19 period.

The findings of this work, supported by recent studies, found that pollution levels—notably $NO_2$ concentrations—decreased significantly in 2020 compared with prepandemic years[22 23 41 42] due to changes in anthropogenic activities. This pattern needs to consider in the context of already declining levels of air pollution in Oxford and exceptional weather in 2020, including a warmer and windier early part of the year, coinciding with the first national lockdown.[22 41] Recent studies by the OxAria team highlighted a clear reduction in traffic volume of 69% and 38% during the first and second lockdowns, respectively, in Oxford, where $NO_2$ emissions from buses and cars reduced by 56% and 77%, respectively.[22 23] While we did not attempt to explore contributions of specific emissions sources to changes in asthma admissions, the statistically significant effect of $NO_2$ and $PM_{2.5}$ in the single-pollutant model was not replicated in the multipollutant model, which indicates that changes in emissions sources may be relevant for determining overall asthma exacerbation risk in this context.

The findings presented in our study regarding the consistently positive association between $NO_2$ and $PM_{2.5}$ and asthma exacerbations rates are consistent with the existing evidence.[43–52] The identified lockdown impact is also broadly consistent with a large-scale analysis of emergency admissions in Scotland and Wales which showed

**Table 2** Incidence rate ratio (IRR) for adult asthma admissions of in response to changes in air pollutant concentrations using the Poisson model

| Pollutants | Single-pollutant model | | | Multipollutant model | | |
|---|---|---|---|---|---|---|
| | IRR | 95% CI | P value | IRR | 95% CI | P value |
| $NO_2$ | 1.040 | 1.009 to 1.072 | 0.012 | 1.026 | 0.987 to 1.067 | 0.188 |
| $PM_{2.5}$ | 1.029 | 1.006 to 1.052 | 0.013 | 1.017 | 0.988 to 1.045 | 0.254 |
| $PM_{10}$ | 1.018 | 0.999 to 1.038 | 0.07 | NA | NA | NA |
| Temperature | 0.98 | 0.950 to 1.020 | 0.340 | 0.99 | 0.960 to 1.030 | 0.620 |
| Relative humidity | 1.01 | 0.990 to 1.020 | 0.466 | 1.00 | 0.990 to 1.020 | 0.826 |

NA, not available; $NO_2$, nitrogen dioxide; $PM_{2.5}$, particulate matter pollution—particles with diameter <2.5 µm.

a 36% reduction in asthma-related hospitalisations for all ages population during the first 18 weeks of 2020,[31] however, the fixed-effect meta-analysis model limited wider generalisability. Davies *et al* proposed multiple possible theories for their observations, including reductions in $NO_2$ and $PM_{2.5}$ that were a national trend, however, unfortunately they were unable to substantiate this proposition as they did not collect concurrent air quality data. Further there exist differences in healthcare access and COVID-19 emergency public health regulation between the devolved nations and England. The first pandemic year, 2020, was also exceptional for the low rates of asthma admissions worldwide,[53] as previously reported in the paediatric population[54–58] and reflected in our own local admissions dataset.

There are other hypotheses for the most likely contributing factors to the reduction in asthma admissions in 2020, including speculation that patients were more vigilant with their asthma prescriptions, and therefore, had improved compliance to preventative inhaled steroid medication.[31] Given the fear about contracting COVID-19 when also diagnosed with a chronic condition[59] and the findings from the RECOVERY trial with adoption of steroids as a treatment for COVID-19,[60] it is plausible that patients improved compliance with their asthma plan, including compliance with any corticosteroid medication, during the pandemic and this contributed to a reduction in admissions during 2020.[31] This explanation has not been widely tested, however, and it is unlikely that compliance continued throughout the entirety of 2020. An alternative theory is that many patients avoided healthcare services either due to fear that they may contract COVID-19 or a moral sense of duty to prevent overwhelming the NHS.[59 61 62] There is weak evidence though for patients with asthma delaying presentation given there was no change in asthma mortality and no increase in acuity of admissions from the studies which looked at this metric.[63 64] This suggests that delays in seeking healthcare assistance were not a reason for the lower admission rates observed here. Furthermore, asthma admission was chosen instead of attendance for the outcome in this study to reduce the chance that the change observed was due to a change in patient behaviour (delay in presentation). Nevertheless, it would have been beneficial for this study and any future studies to include a mechanism for measuring the extent to which a patient delayed presentation. Some studies have suggested that the low asthma admission rate may be due to low person-to-person transmission of other respiratory viruses, however, this is unlikely to completely explain the dramatic reduction in 2020 admissions observed in multiple countries as well as the UK.[31 61 65 66]

## Strengths and limitations

This study has some additional limitations including the ecological, observational design that precludes the establishment of causality. Due to the observational study design it is difficult to conclude that changes in air pollution concentrations were wholly responsible for the reduction in asthma admissions during 2020. During the early pandemic phase several reports suggested that all-cause emergency admissions reduced, as the public responded to the 'protect the NHS' campaign or were too fearful of COVID-19 risks to present to hospital.[61] However, acute asthma is a severe condition which requires urgent hospital management and therefore it is likely that most patients in this study accessed emergency care following an acute attack. Our findings are also consistent with those of Shah *et al*, who identified a statistically significant reduction in asthma patients who experienced an exacerbation during the lockdown period. The aggregated nature of the data collected on asthma admissions and the estimates of the population at risk means that caution must be exercised regarding any inferences at the individual level. Despite a defined population at risk, within a relatively small pollutant area, it is not known whether those admitted with asthma were exposed to the air pollutant concentrations measured. Further, asthma admissions data were aggregated to weekly time intervals due to the need for protection of patient anonymity; therefore, restricting our capability to undertake daily analysis. We were also unable to include further seasonal influences (eg, pollen exposure)[67] in the current study. Finally, we include data for ambient pollutant exposure only and recognise that adult patients with diagnosed asthma are likely to have spent a high duration of time indoors, particularly during lockdown periods. Indoor pollutants can have a significant impact on asthma conditions,[68] and it would be interesting to further investigate the potential impacts of indoor pollution on asthma admissions across the region. Further research is required to understand the respective changes in indoor and ambient pollutant concentrations during the COVID-19 pandemic and to better understand links to chronic disease outcomes among vulnerable patients. In particular, further studies should analyse admissions at the patient level to understand if pollutants have a particular impact on specific subgroups of the population with asthma which could be used to target future public health policies and guidance.

## CONCLUSIONS

This study provides important insights into the effects of changes in ambient air quality on asthma exacerbations in adults during the COVID-19 pandemic in Oxford, UK. The COVID-19 national lockdown measures have led to pronounced changes in air pollutant ($NO_2$, $PM_{10}$ and $PM_{2.5}$) concentrations in Oxford. Our results show that adults with asthma had a significantly lower risk of unplanned hospital admission in 2020 as compared with the prepandemic years (2015–2019). While this study cannot attribute exact causation, our findings suggest that improvements to air quality could reduce asthma healthcare service burden. There is a need to identify which pollutants, sources and locations would be most

beneficial to target with future public health interventions to prevent severe asthma exacerbations requiring hospitalisation in adults.

**Author affiliations**

¹School of Geography Earth and Environmental Sciences, University of Birmingham, Birmingham, UK

²Institute of Applied Health Research, University of Birmingham, Birmingham, UK

³Institute of Microbiology and Infection, University of Birmingham, Birmingham, UK

⁴NHS Oxfordshire Clinical Commissioning Group, Oxford, UK

⁵Department of Engineering Science, University of Oxford, Oxford, UK

⁶Ricardo Energy and Environment, Didcot, UK

⁷Apertum, Oxfordshire, UK

⁸Oxfordshire County Council, Oxford, UK

⁹Oxford City Council, Oxford, UK

**Acknowledgements** We are grateful to staff at Oxfordshire Clinical Commissioning Group, Oxfordshire County Council and Oxford City Council for supporting this work.

**Contributors** SEB, FL, FDP, GNT, BS, TB, SC and GE contributed to funding acquisition for the wider OxAria study. SEB, GLM and GNT conceived and designed the current study. RA and CC acquired linked health data. TB and PA acquired air quality and meteorological data. GLM linked air quality and health data, conducted initial data analysis and drafted the initial report under the supervision of SEB, AS undertook data analysis, wrote the draft article and undertook data visualisation. All authors contributed to editing and critically revising the article and approved the final draft. SEB is guarantor of the work.

**Funding** This research was funded by the Natural Environment Research Council (grant no. NE/V010360/1). Its forerunning pilot project was funded by the National Institute for Health and Care Research (NIHR130095; NIHR Public Health Research).

**Patient and public involvement** Patients and/or the public were involved in the design, or conduct, or reporting, or dissemination plans of this research. Refer to the Methods section for further details.

**Patient consent for publication** Not applicable.

**Ethics approval** Ethical approval for the OxAria study was provided by the University of Birmingham Research Ethics Committee (ERN_20-0831).

**ORCID iDs**

Ajit Singh http://orcid.org/0000-0003-0986-2064

Felix Leach http://orcid.org/0000-0001-6656-2389

Suzanne E Bartington http://orcid.org/0000-0002-8179-7618

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
