## [Reviewer comments · BMJ Open]

ARTICLE DETAILS

TITLE (PROVISIONAL)	Impacts of ambient air quality on acute asthma hospital admissions during the COVID-19 pandemic in Oxford City, UK; a time series study
AUTHORS	Singh, Ajit; Morley, Gabriella L.; Coignet, Cécile; Leach, Felix; Pope, Francis; Neil Thomas, G.; Stacey, Brian; Bush, Tony; Cole, Stuart; Economides, George; Anderson, Ruth; Abreu, Pedro; Bartington, Suzanne

VERSION 1 – REVIEW

REVIEWER	Gonzalez-Barcala, Francisco University Hospital of Santiago de Compostela, Department of respiratory Diseases
REVIEW RETURNED	08-Jan-2023

GENERAL COMMENTS	IMPORTANCE OF THE QUESTION OR SUBJECT STUDIED The study of the relationship between pollutants and asthma admission is a relevant topic for research The objectives are clearly stated ADEQUACY OF APPROACH The experimental design is adequate. However, there are some weak points: It is not clear what hospital admission refers to. There are different types of hospital admissions: emergency room, hospital ward and intensive care unit admissions. Were all of them included or only some of them? There are papers where emergency room admissions are not included. This needs to be clarified. It is mentioned that all patients with primary diagnosis of asthma were included. It is frequent that cases of asthma exacerbation have a different primary diagnosis such as respiratory infection, respiratory failure, bronchitis and so on. Were these diagnoses included? It is stated that patients were included with residential codes OX1-OX4. It is important to clarify where the pollutant samplers were placed and how far they were from these residential areas. How many samplers were available? Is daily data available? The effect of peak values is usually stronger than the effect of mean values (PMID: 23428055) and to this end it seems necessary to analyse the impact for every single day. If these data are not available, this should be cited as a limitation. The statistical treatment is adequate. It is acceptable from an ethical point of view. RESULTS
---

	The results are clearly presented. I think that for every table and figure, abbreviations should be fully spelt as a footnote (e.g., RH in figure 3 and IRR in table 1) It is mentioned that in the multipollutant model the effect of each pollutant was attenuated. However, considering that the effect is not significant, I think that the sentence should be changed to make this non-significant result clearer. DISCUSSION The discussion is relevant The conclusions are supported by the data presented. I have one minor concern with the sentence “The findings presented in our study regarding the consistently positive association between NO2 and PM2.5 and asthma ex” because I do not think that it is so consistent in this study, where the effect in the multipollutant levels is not significant. I think that in the limitations it should be added that indoor pollutants were not considered, although they might have had a relevant impact on the results (PMID: 22704531) Considering that asthma admission rates and pollutant levels are highly variable between regions, it would be interesting to compare the levels of pollutants and the admission rates with other articles. REFERENCES The references are relevant and up-to-date. I think that some references may be useful to improve both the introduction and the discussion. Diesel exhausts particles: Their role in increasing the incidence of asthma. Reviewing the evidence of a causal link. Muñoz X, et al. Sci Total Environ. 2019 Feb 20;652:1129-1138. Influence of pollen level on hospitalizations for asthma. Gonzalez-Barcala FJ, et al. Arch Environ Occup Health. 2013;68(2):66-71. Asthma and COPD exacerbation in relation to outdoor air pollution in the metropolitan area of Berlin, Germany. Hoffmann C, et al. Respir Res. 2022 Mar 20;23(1):64. Asthma admission among children in Hong Kong during the first year of the COVID-19 pandemic. Wong KL, et al. Pediatr Pulmonol. 2022 Dec;57(12):3104-3110. Impact of lockdowns on paediatric asthma hospital presentations over three waves of COVID-19 pandemic. Homaira N, et al. Allergy Asthma Clin Immunol. 2022 Jun 16;18(1):53. Indoor air contaminants and their impact on respiratory pathologies. Carazo Fernández L, et al. Arch Bronconeumol. 2013 Jan;49(1):22-7. GRAMMAR AND STYLE The writing is clear and easy to follow Title The title clearly describes the article ABSTRACT The abstract is adequate and well structured.
--	--

REVIEWER	Sont, Jacob Leids Universitair Medisch Centrum, Bimedical Data Sciences, section Medical Decision Making
-----------------	---

REVIEW RETURNED	16-Mar-2023
-------------

GENERAL COMMENTS	This is an interesting study on the relationship between ambient air pollution exposure and risk of unplanned (emergency) admissions in adults with asthma. Although the paper generally reads well, the research question, analysis and presentation of the results needs improvement. Specific comments There is no specific research question or hypothesis formulated on the potential decrease in air pollution in 2020 as compared to the 5-year average. The authors state that the focus of this paper is on Oxford as a model for smaller towns and cities (line 101). However, they also state that Oxford is an international city with 46000 people coming to the city for work on a daily basis (line 122). Is the air pollution pattern of Oxford indeed representative for other smaller cities? Missing values (line 142) – Missing values in HES data were replaced with the weekly average asthma cases for the study period. Since there are seasonal effects it is better to replace missing values by interpolation based on the adjacent values. Analysis (line 170) – comparison of admission rates should be done by Poisson regression with time period as independent factor. Analysis (line 177) – Since there is a seasonal effect in both the admission rates and air pollution this should also be addressed as possible confounder in the relationship between air pollution and admission rates. The possible seasonal effect should also be addressed in the discussion. Line 214 – A recent study... this sentence should be moved to the discussion Line 219 – corr. coeff. ≥ 0.44 should not be labeled weakly according to DG Altman, Practical Statistics for Medical Research. Line 249 & figure 5 – In contrast to the text the admission rate during September did not remain below the 5-yr average. Tables - Please provide full p-values instead of $p < 0.05$ Table 4 – Please provide estimates for temperature, humidity and season as well Figure 2 – Why is the 2016 data not complete? This seems to have influenced the smooth fit and probably affects the difference between 2020 and 5-year average. Figure 5 – Please indicate the lock-down period Figure 6 – Is similar to figure 2. Could be combined. There are probably too many figures.
--

REVIEWER	Young, S. Stanley CGSTAT
-----------------	-----------------------------

GENERAL COMMENTS

01 Review of Singh V03

Points to cover

Long gap in PM10, PM2.5. Check how it is handled.

Review of Singh

General Comments

This paper examines asthma hospitalizations in the English city of Oxford over the years 2015 to and including the pandemic year of 2020 as possibly induced by three air components, PM10, PM2.5 or NO2, particulate matter of two sizes or Nitrogen Dioxide. Oxford is not a large city, so the number of hospitalization events is small. The authors compensate by analysis of the number of events per month and use monthly average air component levels as predictors.

I focus on things that should improve the paper and help a reader.

The authors cite some of the positive literature. Fine. The Introduction mentions few if any citations of negative studies. There are negative papers. I point to several. Analysis of earlier data from Canada (when air quality was much worse) found no effect (1). Also, data was secured from Los Angeles, US, a city with some of the poorest air quality in the US was negative (2). Both data sets were made publicly available at one time or another. The authors should note there are negative studies. Also, the US EPA exposed human asthmatics to vehicle exhaust gas in chamber experiments and did not induce asthma attacks. [I happen to consider these studies unethical, but who am I.] (3) Finally, the three predictors are very small, too small to be allergens so biological plausibility is in question.

Both Koop and Milloy make their data set available. I have looked at both data sets and I find no association of air components with hospital admissions. It would be useful for people working in this area for these authors to make their analysis data available.

Pollen is often indicted as a cause of hospital admissions. There is a web site in Oxford that lists daily pollen counts, <https://www.kleenex.co.uk › pollen-count › oxford>. They might be willing to provide daily pollen counts, which could be turned into monthly counts.

Certainly, pollen is a lurking variable. Within a given month there could be a spike in an air component or a particular pollen or both. Certainly, pollen is a lurking variable. Within a month there could be a spike in an air component or a particular pollen or both.

This is a good time to comment on the statistical evidence presented, the authors' Table 4, with my computed p-values.

	Data taken for Singh, Table 4 Table Description automatically generated There are five tests in the table, two of which have p-values smaller than 0.05. The statistical evidence is not strong. A speculation: I did notice that admission are up over the months Sept-Jan, which is when students are in Oxford, but in the covid year, not only are admission down, but are down considerable during the Sept-Jan period. Were students not in Oxford for 2020? DATA: It would help the interested reader to have the analysis data set used by the authors, one row per month, asthma count, air components, and covariates, temperature and relative humidity. Some months have missing data. The resulting data set is small by current standards, 6 years, 12 months per year, less than 10 variables. Pollen can induce an asthma attack. There is a group in Oxford that reports daily pollen counts. https://www.kleenex.co.uk › pollen-count › oxford Perhaps they could provide data to examine any relationships of pollen and hospitalizations. Certainly, pollen is a lurking variable. Within a month there could be a spike in an air component or a particular pollen or both. Specific comments Line 33 weakness. No consideration of pollen. Line72ff need reference. Line 79ff reductions in NO2 and many other changes. Also, there might be gains on some things and losses on others. Too many factors change to make sense of things. Line 91 reductions – maybe people just decided not to go to hospital NB mortality did not change. Line 98 reductions might be due to other factors – fear to go to hosp
--	---

	Line 103 goal of study: In this study, a time-series analysis was conducted to investigate the association between ambient air pollution exposure and risk of unplanned (emergency) admissions among adults with asthma living in Oxford, UK. Line 111 Wide time interval might obscure a spike. Line 131 data. Post analysis data set Line 155 Consider using some sort of month by month analysis and eliminate any month with excessive missing air quality or covariate data. Line 155 and figure 2 There is a major gap in the data. There should be recognition and some comment. Also, there is a large spike at 2017 for NO₂, PM₁₀ and PM_{2.5}. Are these results real? Do the results change if they are removed? A sensitivity analysis. Line 163 Usual air components, CO, Oxone and SO₄ are not covered. Authors should comment. Line 165 What are the WHO recommended values? Line 167 At some place show the influence of RH and temp. Should max and mins be used rather than averages? Line 171 PM₁₀ and PM_{2.5} are typically very highly correlated. Why use both? Can you put the problem onto PM_{2.5} rather than PM₁₀? Line 176f The description of Poisson regression is not complete. Were splines used to smooth the background? Was the effect of year removed? Line 186 What about pollen?
--	--

	References Koop, G.; McKittrick, R.; Tole, L. Air pollution, economic activity and respiratory illness: Evidence from Canadian cities, 1974–1994. Environ. Model. Softw. 2010, 25, 873–885. Milloy, S. 2016. Scare Pollution. Bench Press. See Chapter 25. “We have previously completing [sic] a study of [ultrafine particle] exposure in healthy subjects and in health subjects with asthma and there have been no symptoms or airway effects in those studies.” Scare Pollution pg 144 “Based on the 726 emergency room visits for asthma at the VA West Los Angeles Medical Center during the period 2009 to 2011, my data showed that there was no correlation between ozone levels and asthma admissions. I published this result on JunkScience.com” Kindzierski, W.; Young, S.; Meyer, T.; Dunn, J. Evaluation of a Meta-Analysis of Ambient Air Quality as a Risk Factor for Asthma Exacerbation. J. Respir. 2021, 1,173–196. ; https://doi.org/10.3390/jor1030017 Analysis of the Koop et al. data.
--	---

VERSION 1 – AUTHOR RESPONSE

Reviewer: 1

Dr. Francisco Gonzalez-Barcala, University Hospital of Santiago de Compostela Comments to the Author:

IMPORTANCE OF THE QUESTION OR SUBJECT STUDIED The study of the relationship between pollutants and asthma admission is a relevant topic for research. The objectives are clearly stated.

Response: We greatly thank the reviewer for their positive feedback and agree this is a relevant topic.

ADEQUACY OF APPROACH

The experimental design is adequate. However, there are some weak points:

It is not clear what hospital admission refers to. There are different types of hospital admissions: emergency room, hospital ward and intensive care unit admissions. Were all of them included or only some of them? There are papers where emergency room admissions are not included. This needs to be clarified.

Response: Thank you for highlighting this important issue regarding our selection of outcome variable. We can confirm that this relates to all hospital admissions (e.g. not just attendances) during the study period.

It is mentioned that all patients with primary diagnosis of asthma were included. It is frequent that cases of asthma exacerbation have a different primary diagnosis such as respiratory infection, respiratory failure, bronchitis and so on. Were these diagnoses included?

Response: Than you for highlighting this. We can confirm that all patients had a primary diagnosis of asthma. Therefore the data includes those with a range of contributors to asthma exacerbation.

It is stated that patients were included with residential codes OX1-OX4. It is important to clarify where the pollutant samplers were placed and how far they were from these residential areas.

Response: Thank you for highlighting this. The automated air quality monitoring site was an urban background site centrally in the OX1 – OX4 postcode area (covers ~3-5 miles radius). We have now updated sentence on page 7 line 150 in the revised manuscript.

How many samplers were available?

Is daily data available? The effect of peak values is usually stronger than the effect of mean values (PMID: 23428055) and to this end it seems necessary to analyse the impact for every single day. If these data are not available, this should be cited as a limitation.

Response: We can confirm that there is only one automated background site available within the study area that monitors both PM and NO₂ data simultaneously using reference grade instruments. Thank you for recommending additional analysis, however health outcome data (from Oxfordshire Clinical Commissioning Group) was only available for analysis when aggregated to weekly time basis (due to small number suppression), therefore a daily analysis is not possible on this occasion. We have added a sentence to reflect this in the discussion section (page 14, line 360-363).

The statistical treatment is adequate.

It is acceptable from an ethical point of view.

Response: We greatly thank the reviewer for their positive feedback.

RESULTS

The results are clearly presented.

I think that for every table and figure, abbreviations should be fully spelt as a footnote (e.g., RH in figure 3 and IRR in table 1) It is mentioned that in the multipollutant model the effect of each pollutant was attenuated. However, considering that the effect is not significant, I think that the sentence should be changed to make this non-significant result clearer.

Response: Thanks for highlighting this. We have now added the abbreviations in full form as a footnote for the respective figures/tables. We have also changed the sentence to “Within the multi-pollutant model, the effect of each individual pollutant was non-significant.” on page 11, lined 268.

DISCUSSION

The discussion is relevant

The conclusions are supported by the data presented.

I have one minor concern with the sentence “The findings presented in our study regarding the consistently positive association between NO₂ and PM_{2.5} and asthma ex” because I do not think that it is so consistent in this study, where the effect in the multipollutant levels is not significant.

I think that in the limitations it should be added that indoor pollutants were not considered, although they might have had a relevant impact on the results (PMID: 22704531) Considering that asthma admission rates and pollutant levels are highly variable between regions, it would be interesting to compare the levels of pollutants and the admission rates with other articles.

Response: Thanks for highlighting this important issue. We previously included this issue within the limitation section, however we have now expanded this text to provide further clarity for the reader (page 15, lines 366-368) We include data for ambient pollutant exposure only and recognise that adult patients with diagnosed asthma are likely to have spent a high duration of time indoors, particularly during lockdown periods. Indoor pollutants can have a significant impact on asthma conditions, and it would be interesting to further investigate the potential impacts of indoor pollution on asthma admissions across the region. Further research is required to understand the respective changes in indoor and ambient pollutant concentrations during the COVID-19 pandemic and to better understand links to chronic disease outcomes among vulnerable patients.”

REFERENCES

The references are relevant and up-to-date.

I think that some references may be useful to improve both the introduction and the discussion.

Diesel exhausts particles: Their role in increasing the incidence of asthma. Reviewing the evidence of a causal link. Muñoz X, et al. *Sci Total Environ*. 2019 Feb 20; 652:1129-1138.

Influence of pollen level on hospitalizations for asthma. Gonzalez-Barcala FJ, et al. *Arch Environ Occup Health*. 2013;68(2):66-71.

Asthma and COPD exacerbation in relation to outdoor air pollution in the metropolitan area of Berlin, Germany. Hoffmann C, et al. *Respir Res*. 2022 Mar 20;23(1):64.

Asthma admission among children in Hong Kong during the first year of the COVID-19 pandemic. Wong KL, et al. *Pediatr Pulmonol*. 2022 Dec;57(12):3104-3110.

Impact of lockdowns on paediatric asthma hospital presentations over three waves of COVID-19 pandemic. Homaira N, et al. *Allergy Asthma Clin Immunol*. 2022 Jun 16;18(1):53.

Indoor air contaminants and their impact on respiratory pathologies. Carazo Fernández L, et al. *Arch Bronconeumol*. 2013 Jan;49(1):22-7.

Response: Thank you for recommending these references. These citations are now included in the revised manuscript.

GRAMMAR AND STYLE

The writing is clear and easy to follow

Title

The title clearly describes the article

ABSTRACT

The abstract is adequate and well structured.

Reviewer: 2

Dr. Jacob Sont, Leids Universitair Medisch Centrum Comments to the Author:

This is an interesting study on the relationship between ambient air pollution exposure and risk of unplanned (emergency) admissions in adults with asthma. Although the paper generally reads well, the research question, analysis and presentation of the results needs improvement.

Response: We greatly thank the reviewer for their positive feedback.

Specific comments

There is no specific research question or hypothesis formulated on the potential decrease in air pollution in 2020 as compared to the 5-year average.

Response: We have now added a statement on page 8 lines 189-192 "The significant reductions in air pollutant concentrations in 2020 compared to 2015 - 2019 were attributed to Covid-19 national lockdown measures, primarily due to lower traffic emissions.²²"

The authors state that the focus of this paper is on Oxford as a model for smaller towns and cities (line 101). However, they also state that Oxford is an international city with 46000 people coming to the city for work on a daily basis (line 122). Is the air pollution pattern of Oxford indeed representative for other smaller cities?

Response: We agree this is an important point, given Oxford has a relatively high proportion of student population (compared to similar sized UK cities) and inward travel for employment purposes. However overall average pollutant levels (and daily peak values) are broadly comparable to other small and medium sized UK cities, with comparable changes observed during lockdown periods (see Oxford City Council Air Quality Annual Status Report 2020, available at: https://www.oxford.gov.uk/downloads/file/7612/air_quality_annual_status_report_2020).

Missing values (line 142) – Missing values in HES data were replaced with the weekly average asthma cases for the study period. Since there are seasonal effects it is better to replace missing values by interpolation based on the adjacent values.

Response: Thank you for recommending this. Following your suggestion, missing values in HES data have now been replaced by interpolation based on the adjacent values (see page 7, lines 141-144). However, no change was found in the results as a minimal proportion of data values (<4%) were missing.

Analysis (line 170) – comparison of admission rates should be done by Poisson regression with time period as independent factor.

Response: Thank you for recommending this approach. Following your suggestion we have now undertaken the comparisons by Poisson regression analysis. However, there was no notable changes observed in new results.

Analysis (line 177) – Since there is a seasonal effect in both the admission rates and air pollution this should also be addressed as possible confounder in the relationship between air pollution and admission rates. The possible seasonal effect should also addressed in the discussion.

Response: We thank the reviewer for this insightful comment. We consider that seasonality has been adequately accounted for in our methodological approach, given monthly pollutant concentrations are used for exposure assessment purposes. We have added a further sentence to note that additional seasonal effects, such as pollen or allergen influences are not included in our model due to data availability (page 14, line 360-363)

Line 219 – corr. coeff. ≥ 0.44 should not be labeled weakly according to DG Altman, Practical Statistics for Medical Research.

Response: Please could the reviewer provide further clarification regarding our interpretation of correlation coefficients.

Line 249 & figure 5 – In contrast to the text the admission rate during September did not remain below the 5-yr average.

Response: We have now revised this sentence for improved clarity. Page 10, lines 242 – 243 “We also observe a peak in admissions during September, with admission rates nearly matching the five-year average.”

Table 4 – Please provide estimates for temperature, humidity and season as well

Response: Thank you for recommending this. Following your suggestion, we have now included estimates for temperature and relative humidity in Table 4 (now Table 2).

Figure 2 – Why is the 2016 data not complete? This seems to have influenced the smooth fit and probably affects the difference between 2020 and 5-year average.

Response: The Oxford City Council air quality monitoring team were contacted about this issue and we can confirm that during the course of 2016, the urban background monitoring station of Oxford St Ebbe's suffered from a couple of incidents that affected the performance of both PM and NO_x analyser, therefore this period shows missing data.

Figure 5 – Please indicate the lock-down period

Response: Thanks for highlighting this. We have now indicated the lockdown period within the respective figure legend.

Figure 6 – Is similar to figure 2. Could be combined. There are probably too many figures.

Response: Thanks for suggesting this. However, we feel that these should both be retained to enable visual comparisons.

Reviewer: 3

Dr. S. Stanley Young, CGSTAT

Comments to the Author:

Comment on the data gap.

Note the extreme values at ~2017. Comment and consider a sensitivity analysis.

Were other air components considered? CO, SO₄, ozone Consider looking into pollen.

Think about students present/absent relative to admissions.

Asthma is quite complex. It would help interested readers to have a copy of the analysis file you used.

Response: We greatly thank the reviewer for their feedback. We can confirm that this study focuses on PM and NO₂ pollutants due to their known impacts on Asthma; in addition there was no CO or SO₄ data available at the study area.

We will provide all analytical codes upon request.

VERSION 2 – REVIEW

REVIEWER	Sont, Jacob Leids Universitair Medisch Centrum, Bimedical Data Sciences, section Medical Decision Making
REVIEW RETURNED	28-Sep-2023
GENERAL COMMENTS	Thank you for your answers. The paper has been improved.